# Towards an Understanding of Control of Complex Rhythmical “Wavelike” Coordination in Humans

**DOI:** 10.3390/brainsci10040215

**Published:** 2020-04-05

**Authors:** Ross Howard Sanders, Daniel J. Levitin

**Affiliations:** 1School of Health, Faculty of Medicine and Health, The Universirty of Sydney, NSW 2006, Australia; 2Department of Psychology, Faculty of Medicine, McGill University, Montreal, QC H3A 0G4, Canada; daniel.levitin@mcgill.ca

**Keywords:** CPG, swimming, butterfly style, freestyle, coordination, body wave, phase, rhythm, motor control, CNS

## Abstract

How does the human neurophysiological system self-organize to achieve optimal phase relationships among joints and limbs, such as in the composite rhythms of butterfly and front crawl swimming, drumming, or dancing? We conducted a systematic review of literature relating to central nervous system (CNS) control of phase among joint/limbs in continuous rhythmic activities. SCOPUS and Web of Science were searched using keywords “Phase AND Rhythm AND Coordination”. This yielded 1039 matches from which 23 papers were extracted for inclusion based on screening criteria. The empirical evidence arising from in-vivo, fictive, in-vitro, and modelling of neural control in humans, other species, and robots indicates that the control of movement is facilitated and simplified by innervating muscle synergies by way of spinal central pattern generators (CPGs). These typically behave like oscillators enabling stable repetition across cycles of movements. This approach provides a foundation to guide the design of empirical research in human swimming and other limb independent activities. For example, future research could be conducted to explore whether the Saltiel two-layer CPG model to explain locomotion in cats might also explain the complex relationships among the cyclical motions in human swimming.

## 1. Introduction

It has been shown, primarily in bimanual tasks such as tapping [1,2], that the human central nervous system tends to coordinate limbs so that they move in-phase with each other or 180 degrees out of phase [3,4,5,6]. Of these two primary phase relationships, in-phase is the more stable, and, with increasing beat frequency, there is a tendency for a shift in phase to occur from out-of-phase to in-phase. Phase relationships other than in-phase and 180 degrees out-of-phase are typically unstable and, therefore, difficult to sustain [7], resulting in high variability of relative phase and spontaneous unintended shifts to a more stable phase relationship. In a review of bimanual coordination [8], it was reported that only simple bimanual timing ratios such as 1:1 can be performed without extensive practice while other ratios (e.g., 1:2, 2:3, 3:5) are significantly more difficult to perform. These other ratios are most noticeably apparent in musical settings, especially involving indigenous music, and humans can achieve expertise in reproducing them [9,10].

Given the difficulty of sustaining complex rhythms in bimanual tasks, it is surprising that coordinated movements in sports exhibit phase relationships among body segments that are not simply in-phase or 180 degrees out-of-phase. These phase relationships are developed to optimize the transfer of energy through the mechanical system and success in the task. For example, skilled swimmers learn to adjust the phase of various actions to optimize speed within the physiological constraints. Sanders et al. [11] found that the vertical oscillations of body parts that culminate in a highly propulsive kick in butterfly swimming are sequenced so that a sinusoidal two-beat “body wave” travels caudally from head to feet. A second four-beat body wave travels caudally from the hips to the feet. However, this optimal body wave is achieved only with complex phase relations.

Table 1 shows the phase differences between the oscillations of the body parts, the velocity of the travelling two-beat body wave and the correlation between the speed of the body wave and the center of mass velocity of elite United States butterfly swimmers.

Table 2 shows the phases of the two waves for a typical national-level butterfly swimmer. The phasing of the two waves results in a strong upward kick (which generates torque to raise the upper body) followed by a strong downward kick (to lower the upper body). This then reduces the effort required to raise the upper body and frees the arm action to generate propulsion rather than lift. In this manner, energy is retained in the system and recycled effectively between stroke cycles. By optimizing the phases of the two waves, skilled butterfly swimmers can swim 200 m butterfly (current world record = 1 min 50.73 s) almost as fast as they can swim 200 m front crawl (current world record = 1 min 42.00 s). Interestingly, the speed of the travelling two-beat wave relative to the body is slightly faster than the forward speed of the swimmer and is similar to that found in marine animals such as whales and dolphins [12]. From a hydrodynamic perspective, a small difference between the speed of the body wave and the speed of progression of the swimming animal is indicative of efficient swimming.

Progression of body waves has also been investigated in front crawl swimming [13,14] in which “torsional” waves progressing caudally were examined. A two-beat torsional body wave, initiated through the rolling of the upper body about its longitudinal axis, supplemented a six-beat kicking pattern represented by a six-beat travelling torsional body wave. It was shown that the speed of the torsional six-beat wave relative to the body was closer to the speed of progression of the swimmer (approximately 1.8 metres per second) for highly skilled swimmers than less skilled swimmers (Table 3). Thus, it appears that increasing skill in flutter kicking involves increasing the phase differences to produce moderate velocities of wave travel. It is also noteworthy that the skilled swimmers have an accelerating wave from hip to ankle achieved by increasing the phase difference from hip to knee so that the hip to knee velocity is less than the knee to ankle velocity.

Becoming skilled in front crawl swimming requires the development of appropriate phase relationships between the arm actions, a two-beat body roll, breathing actions, and the six-beat kick. The relationship between the actions of right and left upper limbs changes continuously throughout the front crawl stroke cycle and varies according to the constraints related to the ability of the swimmer to supply, through the aerobic and anaerobic metabolic processes, energy for mechanical work. This is reflected in differences in the “index of coordination” [15] between sprint and distance swimming. Nevertheless, this complex movement pattern remains consistent across stroke cycles [16].

Thus, humans optimize performance in butterfly and front crawl swimming through the development of complex yet stable phase relationships among joint actions. However, given that complex phase relationships are extremely difficult to achieve in seemingly simpler tasks such as bimanual tapping, the question of how the central nervous system (CNS) might achieve complex rhythms among several multi limb actions needs to be addressed. The emergent kinematics of skilled butterfly swimming and front crawl swimming have been shown to be rhythmical by virtue of being composed of sinusoidal vertical undulations in butterfly swimming, and sinusoidal torsional rotations in front crawl swimming. The phase relations among the rhythms have been explained in terms of optimizing the kinetics, energetics, and hydrodynamics [13]. However, understanding of the neural control of the movements is lacking. To provide insights and possible explanations of how the human neurophysiological system might be organized to achieve the optimal phase relationships among the composite rhythms, we conducted a systematic review of literature relating to central nervous system control of phase among joint/limbs in continuous rhythmic activities.

## 2. Materials and Methods

To address the research question, a systematic search of the existing literature was conducted using the combined keywords “Phase AND Rhythm AND Coordination”. The rationale underpinning this choice was directly related to the task of explaining how the coordination of rhythmic motion is achieved in complex cyclical activities with phase relationships among the rhythmic motions being stable at phase angles other than 0 or 180 degrees. Two major databases that draw on subsidiary databases were searched—SCOPUS and Web of Science. In Web of Science, the “All data bases” option was selected. Given the rapid advancement in experimental approaches and mathematical models of CPGs, paper published in the period 1999 to the time of writing (December 31, 2019) were reviewed. However, pertinent earlier articles that were cited as foundational to the recent work were included when necessary in the introduction or during the interpretation of the contribution of each paper in the discussion. Also, from around the year 1999, there has been a consistent “critical mass” of more than 20 papers per annum identified through the chosen keywords (Figure 1). The selected papers were all peer-reviewed journal articles with impact factors over 1.5. Conference proceedings papers and dissertations were not considered.

Two categories of criteria were applied in filtering the papers identified by the keyword search—quality and relevance. To meet the quality criteria (Figure 2) a paper needed to report empirical data with a rigor of methods and replicability evident from the details of the experimental procedures, as determined by the first author and checked by the second author. Given that the range of experimental approaches included in vivo, in vitro, and mathematical modelling, those general, rather than specific, criteria were applied to ensure quality.

To be considered relevant, a paper needed to provide evidence for mechanisms of control that could contribute to explaining how swimmers might maintain stable relationships among oscillations of body parts in swimming. The criteria for inclusion are shown in Figure 3. The process was conducted sequentially by assessing relevance by title, then by abstract, and then by reading the full papers to further assess relevance and quality.

Only one paper that had progressed to a reading of the full paper was deemed low quality. That paper also had limited relevance. Many rigorous papers were filtered during either the abstract assessment (21 papers) or during the reading of the full text (16 papers). Common reasons were (1) Focus on methods of understanding or treating dysfunctional rhythmical movement rather than functional and normally coordinated rhythmical movement; (2) Focus on output behavior or EMG rather than the central nervous system control; (3) Focus on instability rather than maintenance of stable and sustainable movement; (4) Focus on entrainment of oscillatory behavior rather than maintenance of phase relationships other than in-phase or 180 degrees out of phase; (5) Focus on single joint behavior or coupling of a limited number of joints (often with an injury focus). The papers that were filtered out were checked by the second author to ensure agreement with regard to inadequate relevance.

## 3. Results

Figure 4 shows the results of the systematic search of the literature. Table 4 shows the articles reviewed and brief information regarding the purpose, the sources of data, the species studied or modelled, the name of the journal and its most recently available impact factor.

## 4. Discussion

Knowledge of how the central nervous system operates has developed over time. Here we discuss the contributions of the reviewed papers in loosely chronological order while allowing some flexibility to maintain flow and integrity of sub-themes.

Researchers have sought to understand how the nervous system coordinates body segments by using phases that enable efficient motion. Mathematical modelling has been common as a means of assessing whether observed characteristics of locomotion of various organisms can be replicated with a small number of model parameters. Calvitti and Beer [17] studied locomotion of a stick insect (Carausius morosus) by applying the mathematical model of Cruse [39] in which the timing of the protraction and retraction of a “receiving leg” depends on the state of the “sending leg”. This means that the movement of the receiving leg is “phase-locked” to the “sending leg”. Simulations showed that the phase relationship is “phase compressed” rather than “phase-locked”, i.e., there is a capacity for some variability from cycle to cycle at a given average walking speed and that stability is maintained by other mechanisms, including response to feedback from the leg oscillators. The authors stated that although the model was designed for arthropods such as stick insects, it could also be applied to other species, including cats. The concept of relevance in the current review is that the timing of the motion of successive body segments or joints could be pre-determined by simple parameterization of the mechanical system by the CNS, thereby simplifying the control of the movement sequence and enabling expedient fine-tuning of it to optimize performance. In the case of butterfly swimming, the phasing of the four-beat waveform commencing at the hips might be linked to the two-beat waveform emanating from the vertical oscillations of the upper trunk.

Saltzman and Byrd [18] developed a non-linear dynamical systems model to explore the phase relationships between gestures in speech, comprising phase relationships among speech articulators primarily but also with potential application to the coordination of limbs. Their “extended” model showed that self-organization of phase relationships between gestures can be achieved through the attractor states of coupled oscillators. This allows the variability of the phase relationship required in speech, but within a defined range (phase window). Importantly, a target phase relationship can be set, that is, not fixed, to the usual stable phases such as 0 degrees or 180 degrees. The implication for swimming is that target phase relationships among participating actions may be set by the CPGs to optimize performance within the various constraints, including the physiological constraints. This would enable phase relationships to adapt in response to changing task demands and would appear to fit well with the differences in coordination observed with swimming event distances [15,40,41]. Another outcome of the modelling by Saltzman and Byrd was that, where there are oscillators of different frequencies, coupling is stronger to the lower frequency than the higher frequency. Thus, if the CPGs in human swimming work in a manner resembling the Saltzman and Byrd model, the relative phase of the actions in front crawl and butterfly swimming would be linked to the phase of the whole stroke cycle rather than to the phase of the higher frequency oscillations. In front crawl, this would be the two-beat sinusoidal body roll about the long axis rather than the six-beat kicking action. In butterfly swimming, the four-beat travelling wave from the hips would couple to the two-beat travelling body wave.

There is evidence that in situations in which discrete movements are combined with oscillatory movements, as in the case of butterfly and front crawl swimming, the onset of the discrete movement is constrained to a narrow phase of the oscillatory movement [20,42]). Sternad and Dean [20] hypothesized that discrete and rhythmic movements tend to synchronize. In a “table cleaning” task, in which linear oscillatory movements of the hand centered on one location had to be transferred on command to a different target location, predominantly by shoulder abduction, there was a systematic phase advance when moving to the new target. Also, the initiation of the shoulder motion enabling the translation of the hand to the new target was constrained to a preferred phase of the elbow oscillation. Thus, it may be, for example, that the commencement of backward hand movement to make the “catch” in swimming is constrained to the appropriate phase of the body waves in swimming, namely, the phase of trunk oscillation in butterfly and the phase of the shoulder roll in front crawl swimming.

There is evidence that in walking, adjacent joints can be controlled with stable phase relationships. For example, Emmerik et al. [21] found that the oscillations of the pelvis and trunk during walking were neither in-phase nor 180 degrees out-of-phase, and the phase relationship changed with walking speed. Thus, the system exhibited both flexibility and stability in the walking gait of healthy subjects. The authors also provided evidence of stable coordination of oscillations among subsystems, such as locomotor and respiratory systems, and reported that coupling between CPGs for locomotor and respiratory rhythms has been identified at the level of the spinal cord in the spinal rabbit [43]. This supports the idea that the subsystems in swimming may have primary control by spinal CPGs with stable phase relationships that may change with swimming cadence. However, it remains unclear whether this organization can be extended to swimming in which subsystems are operating at different cycle frequencies. As described in the introduction to this paper, the phase relationships between the travelling two-beat and four-beat cephalo-caudal waves in butterfly are critical to efficient swimming and performance. Similarly, in front crawl swimming, the relationships between the torsional oscillations comprising subsystems of shoulder roll (two-beat), arm actions (four-beat), and the two- and six-beat travelling waves of the pelvis to lower limbs, differ among skill levels.

While it is possible that complex rhythmical movements are coordinated through the innervation of muscles by CPGs at the spinal level, it is also known that involvement of the brain is required to maintain stable coordination in humans. By investigating the activity of the brain during finger tapping, Dhamala et al. [19] provided insights into the relationship between brain activity and the rate and complexity of the rhythms. The primary motor cortex, premotor cortex, auditory cortex, basal ganglia, thalamus, and the cerebellum were more active during the task than during the rest periods, and the activity level correlated with tapping rate. Activity in the cerebellum increased with the increasing complexity of the rhythm, along with increased activity in the thalamus as the pathway between the cerebellum and cerebral cortex. Because mammalian movement is typically the result of motivational states, and these are generally controlled by emotions, it has been suggested that the cerebellum is also involved in emotional experience, a finding supported by the work of Schmahmann [44], who has drawn out the connectivity between regions of the cerebellum and the cerebral cortex, particularly the frontal lobes and limbic system. This emphasized the role of the cerebellum in the temporal coordination of actions. Other evidence of the influence of the brain in control of rhythmic movement emerged from the study of Ford et al. [22], who found that an auditory cue was effective in re-establishing walking rhythms among subjects whose coordination was affected by a cerebro-vascular accident.

The role of the motor cortex in the control of gait in cats was investigated by Drew et al. [23]. They proposed that adjustments to gait, for example, stepping over objects, was achieved by subpopulations of cortical neurons that modify the activity of the pattern generators involved in the sequential innervation of muscle synergies. Different limb trajectories could be produced by differentially modifying the activity in each synergy. The synergies involve muscles of several different joints and muscles could be involved in more than one synergy. This organization is pertinent in understanding how movements with amplitudes that are a summation of rhythms of different frequencies and phases may be produced.

Increasingly, models have been developed that include the interaction between CPGs and higher centers. Kozlov et al. [24] tested a biologically realistic CPG model comprising 6000 E neurons projecting ipsilaterally and 4000 I neurons projecting contralaterally, enabling replication of the observed operation of the spinal CPGs in which the intersegmental phase lag is flexibly set to produce the travelling body wave of Lamprey in both forward and backward swimming. The model also considered the contribution of the basal ganglia and brainstem in the control of the CPGs. Positive and negative phase lags initiated at the rostral end of the network control backward and forward swimming motions, respectively, while turning is achieved by bilaterally asymmetrical activation levels. Figure 5 displays the connectivity of principal areas responsible for the motor control and coordination of complex rhythmic activities.

It has been recognized that the neural control of rhythmic movement is highly influenced by the morphological and stiffness characteristics of the body segments. For example, the robotic elbow and leg system developed by Pitti et al. [45] included control elements that mimicked the neuromodulators that fine-tune the CPGs in a biological system. Complexity was reduced, and energetic efficiency increased by phase synchronization that matches the internal dynamics to the dynamics of the mechanical system. A higher control may switch between different muscle synergies to change the stiffness of the muscles to adjust to disturbances or changing demands within a gait cycle. For example, the stiffness of the muscles may be increased during the stance phase and then reduced during the swing phase. This is very interesting with respect to the control of muscle synergies in swimming in which the torques applied at the upper limb joints must change markedly between the push/pull and recovery phases of the stroke in both front crawl and butterfly. The morphology of the body is also interesting in terms of the inertia and consequent natural frequencies of oscillation in response to stiffness modulated by the muscles. It is also pertinent with respect to the amplitude of the oscillations produced in the hips in butterfly swimming, which enables the transfer of energy from the trunk to the lower limbs culminating in propulsion [13].

Some indirect insights into how the phase of movements may be set and maintained come from the work of Ledberg and Robbe [25]. The authors suggested that the sensory feedback of oscillatory motions during locomotion influences the output of the hippocampus to contribute to spatial awareness. In running rats, the theta frequency oscillations of the hippocampal local field (LFP) were of a similar frequency to the frequency of the oscillation of the head. Both the amplitude and frequency of the head oscillations increased with running speed. A finding that the authors reported as “unexpected” was that the amplitude of the hippocampal theta rhythm was related to the phase lag between the head movements and the LFP oscillations. Further, there was little evidence of phase locking. Although the authors were careful not to speculate on the role of the phase lags between the theta waves and head oscillations, it is tempting to propose that the changing phase provides information, via the theta signal amplitude, about the spatial-temporal status of oscillating body parts. If that was then compared to a spatiotemporal reference of what is expected at that stage of the movement, then this may enable downward signals to modulate the CPGs to maintain the optimal rhythm and spatio-temporal relationships among oscillating body parts. In this manner, the phase relationships among the oscillating body parts that enable optimum performance might be maintained.

By setting different split treadmill speeds for the forelimbs and hindlimbs of adult cats, Thibaudier et al. [27] showed that coordination between the fore and hind limbs is bidirectional. That is, the speed set for the fore-limbs influenced the duration of the stance and swing phases of the hindlimbs and the speed set for the hindlimbs influenced the phase duration of the forelimbs. However, the respective influences on cadence were asymmetrical. When the speed of the hindlimb treadmill was faster than the speed of the forelimb treadmill, the forelimbs adjusted to maintain a 1:1 match with the cadence of the hind limbs. In contrast, when the speed of the forelimb treadmill was faster than the hindlimb treadmill, the 1:1 rhythm broke down as the forelimbs adopted a shorter cycle (higher frequency of cadence) and shorter phase durations. These results suggested that the CPGs for the forelimbs imposed their rhythm on the hindlimb CPGs. The authors contrasted these findings with those of Juvin et al. [46] who showed in the in-vitro isolated neonatal rat spinal cord that hindlimb CPGs imposed their rhythm on forelimb CPGs. Consequently, it is interesting to contemplate how the CPGs controlling the rhythms of the arms in swimming might influence the rhythms of the legs and, vice versa, how the CPGs of the legs might influence the rhythms of the CPGs controlling the arms. In front crawl swimming, an increasing cadence of the upper limbs as one progresses from distance to sprint pace invokes an increase in the frequency of kicking so that the kicking beats are completed in correspondence with the upper limb cycle. In doing so, the kicking pattern typically changes from a two-beat to a six-beat pattern. Then, as the arm cycling rate increases, so does the rate of kicking to complete the six beats within the cycle. The cycling rate and phase pattern of the upper limbs are also influenced by the physiological constraints, including aerobic capacity and strength. Thus, one could propose a hierarchy of influence of motor cortex, upper limb CPGs and lower limb CPGs with adjustments at all levels based on sensory feedback.

In butterfly swimming, the upper body is raised through the combined actions of the out-sweep of the hands and up-beat of the kick. Both actions produce torques to raise the upper body to input energy to the system that is transferred caudally by the two-beat body wave [13]. The four-beat body wave emanating from the hips must be timed with an appropriate phase, as shown by experimental data and simulations [13], to produce a four-beat kick characterized by a strong up-beat and strong down-beat. The difference in amplitude of the two up-beats and two down-beats arises from the summation of the two-beat and four-beat body waves and is dependent on their phase relationship. Given the mutual reliance of the upper and lower body actions and the critical importance of the phase relationships among them, it appears likely that there is a bidirectional influence of the upper- and lower-limb CPGs to maintain an effective and economical movement sequence that is stable from cycle to cycle.

There is a paucity of studies in which the ability to maintain a fixed phase relationship between movements at a frequency that is an integer multiple of the fundamental frequency of a movement cycle. Most have investigated the stability of 180 degrees out-of-phase of movements that are at the same frequency as the reference movement. In these experiments, the phase has become unstable and shifted to in-phase with increasing speed [3,5,6]. Snapp-Childs et al. [26] conducted such a study to test the three hypotheses of the dynamic model of Bingham [47]. These were (1) Being able to produce stable coordinative movements is a function of the ability to perceive relative phase, (2) the information to perceive relative phase is the relative direction of motion, and (3) the ability to resolve this information is conditioned by relative speed. Participants were instructed to control a joystick to move a dot on the screen at 180 degrees to the oscillation displayed on the screen. All three hypotheses were supported and, notably, phase-switching from 180 degrees to 0 degrees occurred at oscillation frequencies of about 1.25 to 1.5 Hz. Given these results, it is intriguing that in both front crawl and butterfly swimming, complex phase relationships that are essential to optimize performance are maintained across cycles and that these phase relationships are maintained for body and segmental rotations of different oscillation frequencies.

Axial progression of body waves has been shown in limbless marine animals and in-vitro spines of tetrapods. For example, Ryczko et al. [29] used video-based kinematic analysis and indwelling EMG of the axial musculature to establish that body waves progress along the bodies of freely moving salamanders and that the nature of the waves varies according to the task. When swimming, or backward stepping, waves travelled posteriorly and corresponded to the propagation of waves of EMG that travelled at a faster rate than the kinematic wave. In forward stepping, the waves were described as “standing waves” characterized by a small phase lag between segments. In-vitro investigation of the isolated mid-trunk cord showed that rhythmic motor patterns could be generated. The authors suggested that the organization of rhythmic motions in the salamander might be an evolutionary extension of an axial network of limbless vertebrates like lamprey to include more recently evolved limb networks. Interestingly, when the mid-trunk cord is isolated from the limbs, the frequency of wave propagation is higher than when the limbs are involved. This suggests a link between the generation of rhythms in the limbs and the spinal rhythm generators. Based on their observations, the authors proposed that there is a local coupling between the limbs and the CPG network. The authors recognized that descending signals from higher levels of the nervous system, in combination with sensory feedback, would enable increased control and versatility of rhythmic behaviors. 

In recent years further development of computer models to replicate movement behaviors has enabled fresh insights into how movement may be controlled parsimoniously by neurological systems. For example, Zhang et al. [23] used computational fluid dynamics in conjunction with a neural model of CPG circuits to show that a locked phase difference between adjacent crayfish swimmerets of approximately 0.25 of the period is more hydrodynamically efficient than being in-phase (0) or 0.75 of the cycle period regardless of speed, size of the crayfish, or swimmeret cycling frequency. Spardy and Lewis [36] have subsequently extended the model of Zhang et al. for control of crayfish swimmerets to include the effects of neighbouring neural circuits that are beyond the nearest neighbour. While the nearest neighbour circuits have the dominant effect of setting the phase difference to 25%, the model showed that the longer neighbours can also have a small effect. The authors posited that this may reduce the phase difference between neighbouring swimmerets towards a phase difference that is even more hydrodynamically efficient.

The modelling of Zhang et al. [28] and Spardy and Lewis [36] is interesting in relation to human swimming for two reasons: it illustrates control by CPGs of separate appendages that is neither in-phase or 180 degrees-out-of-phase, and it shows that a stable pattern has evolved to optimize performance. Adjustment of phase between adjacent joints through this mechanism is pertinent given the need to establish optimal phase differences between remote joints in human swimming. For example, the phase relationships between the upper and lower limb cycles are important to optimize the torsional (rolling) rhythms in front crawl and the undulating rhythms in butterfly swimming. In particular, the timing of the actions of the hands must be tuned to the phase of the kick and vice versa. However, there are several major differences when seeking to apply the neural organization of crayfish swimming to human swimming. First, the cycling frequency of the appendages are the same in crayfish swimming but not in human swimming. Second, the optimal phase relationship has emerged through evolution and is transmitted genetically, whereas, in human swimming, the optimal phase relationships have been learned by individuals and are flexible. Third, reflexive proprioceptive feedback can enable swimmerets to increase the motor drive to adjust to increased loads, but not to change the coordination between swimmerets, whereas feedback in human swimmers enables changes in coordination and learning to optimize future performance.

Harischandra et al. [30] demonstrated that models of CPGs, comprising phase-coupled Hopf oscillators to generate rhythm (RG) and “pattern formation networks” (PF) for capturing the frequency and phase characteristics of the oscillations of the joint, could replicate the elliptical searching behavior of stick insect antennas. The efficient elliptical searching behavior results from the distal scape-pedical joint having a phase lead of 10–30 degrees relative to the proximal head-scape joint. This stable phase relationship was maintained with a small number of parameters in the model. Thus, it is possible that specific stable functional phase differences other than 180 or 0 degrees can be maintained between adjacent limbs using CPGs with relatively simple parameterization.

Models of neural control of mammals have also been developed. Hunt et al. [31] reproduced the walking gaits of rats by innervating realistic muscle and joint representations of the individual limbs. The model comprises neurons and synapses with separate rhythm generators for flexion and extension of each limb. An important finding was that the phases among the limbs could be readily adjusted by activation of the elbow extensor motor neuron with associated inhibition of the hip flexor motor neuron. Similarly, Danner et al. [32] produced a model that closely replicates the gait of mice. The model comprised a spinal rhythm generator for each limb with interactions between the limbs via left-right and fore-hind commissural excitatory and inhibitory neurons. Realistic changes in phase among the limbs and transitions from walk to trot and bound were induced in response to increasing drive from the brainstem.

The concept arising from these models of mammalian locomotion is that changes in phase associated with different gait patterns can be achieved through relatively simple adjustment of the amplitude of the signals innervating muscle groups. Thus, it may be that adjusting the phase among body parts to optimize performance in human locomotion is achieved simply by changing the strength of the signal innervating specific muscle synergies. In that vein, Chen et al. [34] showed that, based on EMG data from muscles of the forelimbs and hindlimbs, a muscle synergy for each of the stance and swing phases of each limb was consistent in structure among subjects, and across speeds, of crawling humans. They proposed that this indicated that humans share a “common underlying muscle control mechanism for crawling”. The phase between contralateral limbs remained consistent. However, the recruitment levels, durations, and phases of the muscle synergies changed with crawling speed. The phase lag between ipsilateral limbs also varied across speeds and differed among subjects. The authors proposed that these results were in alignment with control by Rybak et al.’s [48] two-level CPG comprising a half-center “rhythm generator” (RG) and a “pattern formation” (PF) circuit. In that model, the PF level considers afferent feedback and excites or inhibits muscle synergies, which then modify the rhythm generated at RG level to influence the duration of the flexor and extensor phases while maintaining a stable rhythm.

This raises the question of whether proprioceptive feedback from the undulating body parts of the butterfly swimmer might be used to maintain a rhythm at the RG level, which influences the excitation and inhibition of motor synergies to create the sequenced muscle activity to maintain the functional body wave. The next question is whether there is a second RG and PF system for the four-beat pattern from the hips to the ankles. Then, the question arises as to how the two systems interact to create the appropriate phase relationship which has been shown to be essential for energetic and hydrodynamic optimization. Extending to the front crawl situation, the question arises as to whether there are separate rhythm generators for each of body roll, arm action, and kick. If so, how do they interact to optimize the phase relationships between them?

Insights into these questions emerge from the development of models used in robotics. For example, Dutta et al. [38] have shown that the interaction of coupled oscillators can generate a range of gait patterns with synchronized limb movement with a small number of control parameters. The robot control system is inspired by the biological human gait system in which signal strength (gate voltage) is controlled by spinal CPGs modulated by sensory feedback and reciprocal inhibition from the musculoskeletal system and with input from brain centers, including cerebral cortex, cerebellum, basal ganglia, and mesencephalic locomotor region. The robot control replica comprises central pattern generator hardware (spinal cord CPGs), actuators (muscles), “environmental sensors” (sensory feedback), and higher control (brain centers). The CPG hardware comprises capacitively coupled Vanadium Dioxide nano-oscillators, which enable stable limit–cycle oscillations and programmable phase-patterns. These are influenced by the feedback signals to enable the system to cope with perturbations such as changing terrains and obstacles. The gait pattern is determined by the phase differences among the oscillators in response to resistor-controlled voltages of the oscillators. Consequently, different stable gait patterns of quadrupeds such as walk, trot and gallop, can emerge. Transitions between gaits can be achieved via phase shifts induced by differences in voltages that affect the natural frequencies of the individual oscillators. Of interest in relation to the current problem of controlling the undulations in butterfly swimming is that the system is versatile and yet parsimonious with respect to control mechanisms. That is, it comprises CPGs that could maintain rhythmic motion with a small number of oscillators at spinal level but with flexible modulation in response to sensory feedback and input from higher brain centers. Importantly, the oscillators can operate with deliberate phase differences to obtain the desired movement pattern and temporal relationships between actuators. In the case of butterfly and front crawl swimming, these could be the actions of muscles of the shoulders, hips, knees, and ankles that are timed to produce travelling body waves of frequencies that optimize swimming speed within the physiological constraints.

The arm actions in both butterfly and front crawl swimming are complex. While they need to be timed appropriately with the rhythmical whole-body motions and travelling body waves, and are cyclical, there are spatiotemporal targets in their movements. This means that they may be considered as discrete movements with temporal relationships within the overarching whole-body rhythms dictated by the body waves, i.e., the two- and four-beat waves traveling caudally in the butterfly and the two- and six-beat torsional waves travelling caudally from the hips to ankles in the front crawl. The targets of the upper limb actions in both strokes include the entry and exit points of the hands and the lateral excursions of the out-sweeps and in-sweeps. These discrete movements are constrained temporally because the forces produced by the upper limb actions are related to the speed of the motion. This means that their duration requires some independence from the body wave durations to enable swimmers to adjust to the physiological constraints associated with different event distances. This flexibility is achieved by coordinating the commencement of the propulsive actions with the body wave rhythms. That is done by lengthening the period of recovery, entry, a period between entry and “catch” and is evident quantitatively in an “index of coordination” [15]. The question arising is whether the frequency and timing of the body rhythms are set to the timing of the upper limb motions or, conversely, the upper limb motions are set to fit the body rhythms. Based on the Dutta et al. [38] model, one could hypothesize that the rhythms of the body waves are controlled by spinal CPGs with modulation by higher centers in response to the sensory feedback of physiological status/fatigue and by proprioceptive feedback indicating the positions of the joints and end effectors (hands and feet).

An important consideration with respect to maintaining optimal movement patterns in swimming is that the actions of the arms are decoupled with respect to the rhythmic motions of the body and lower limbs. In front crawl swimming, the predominant rhythm is the two-beat rhythm associated with the body roll. The motion of the lower limbs has a two-beat rhythm aligning with the body roll and a six-beat rhythm aligning with the kick [14]. Although the hands must complete their cycle in the same duration as the two-beat rolling and six-beat kicking rhythms, their angular motion relative to time within the cycle is not sinusoidal. There are phases (periods rather than “phase”, as referred to in “phase angle”) within the upper limb cycle involving both linear and angular accelerations of the hand to optimize propulsion and body alignment. For example, the pull phase is faster than the recovery phase. This requires that the relative phase (angular phase) of the upper limbs is not coupled to the phases of the body roll and kick. Also, the relative phase of the right and left upper limbs must be free to vary throughout the stroke cycle in front crawl swimming. In butterfly swimming, bilaterally symmetrical actions reduce the complexity as the right and left upper limbs can move in synchrony. Nevertheless, in both front crawl and butterfly, target events of the upper limbs must correspond to optimal phases (angular phase of the body waves) of the rhythms of the other body parts, i.e., the two-beat body roll in front crawl and the two-beat and four-beat body waves in butterfly [13]. Thus, although the actions of the upper limbs are decoupled from the rhythmical body actions in the sense that the phase relationship varies throughout the cycle, there may be enforced coupling between the rhythms of other body parts and the timing of the initiation and completion of the discrete actions comprising the arm strokes.

Amado et al. [33] provided evidence that actions in complex activities involving rhythmical motions may be decoupled to maintain performance. Expert marching band members performed three different drum rhythms (1:1, 2:3, and 2:3 fast) in three different postural conditions (seated, two-legged standing, and one-legged standing. The rate of postural sway increased from the two-legged to one-legged condition and was influenced by the complexity of the drumbeat rhythm. However, the posture did not affect the ability of the musicians to maintain the drum rhythm. Further, the coupling between postural movement and the drumbeat rhythm decreased with increasing difficulty of the postural conditions. This decoupling was interpreted as being functional, that is, to enable the drum beat rhythm to be maintained. Additional evidence for decoupling of rhythmical upper limb actions from other body movement rhythms emerged from the study of Qi et al. [37] in which non-musicians performed rhythmic finger tapping in combination with self-paced walking, given-paced walking, alternative bilateral heel tapping, and heel tapping with one foot ipsilateral to the tapping finger. It was found that the walking conditions were independent of the finger tapping, but the heel tapping was not. The authors suggested that finger tapping and walking are controlled by separate locations of the spinal neural control centers.

In both butterfly swimming and front crawl swimming, the sequencing of joint rotations leads to a wavelike transmission of motion caudally as indicated by, and determined by, the phase differences between adjacent body parts [11,14]. While it is useful to compare the control of these wavelike motions to that of other species, it is also necessary to recognize that control of cephalo-caudal wave propagation by CPGs to produce undulating waves along the spine of animals, such as lamprey [24,49], and salamander [29], may differ from control of wavelike actions in which the wave motion is produced by appropriate timing of limb joint flexions and extensions rather than sequential innervation of muscles flexing and extending adjacent vertebral joints. On the other hand, evolution may have endowed vestiges of control mechanisms and organisation. Thus, CPG control in human motion may retain commonalities through the evolutionary process so that elements of the organizations of CPGs of fish are also evident in limbless reptiles, limbed reptiles, crustaceans, amphibians, quadruped mammals, and primates. Indeed this review has revealed organizational models with commonalities among species, including lamprey [24], salamander [29] and crayfish [28,36], and in mammals such as rats [25,31], mice [32] and cats [23,27]. In this vein, the comparison of the control of the locomotion of frogs and cats by Saltiel et al. [35] is pertinent. They stated that while the concept of a longitudinal travelling wave has been well established for limbless vertebrates such as the lamprey or zebrafish, it has been proposed as a mechanism for locomotion of vertebrates with limbs only recently. The tenor of their review, which included results from their experimentation, was that locomotion in the frog was controlled by spinal CPGs with a rostro-caudal sequencing of muscle synergies. While the temporal patterning in the frog reflected a travelling wave organization, the temporal patterning in the cat, characterized by bursts of activation of muscles such as the long head of the tricep and cleidobrachialis controlling retraction and protraction of the shoulder, suggested a “temporal grid”. However, rather than the “temporal grid” being a distinct CPG organization and independent of the travelling wave organization, Saltiel et al. proposed that the CPGs comprise two layers: a “pattern formation layer” (PF) and a “travelling wave layer” (TW). Applied to swimming, this might mean that the activation to produce the desired movement characteristics at the joints would be controlled by the PF layer of the CPG, but the sequencing to produce the wavelike coordination between the body segments would be controlled by the TW layer of the CPG. Input of a theta oscillation from hippocampus/medial entorhinal cortex (mEC) circuitry, represented as a travelling wave rhythm, would control the cycle rate and speed of locomotion.

### Limitations

The review was limited to papers from 1999 to present for the reasons outlined in the method section. Although some foundational papers published prior to that period are identified when discussing the selected papers, it is acknowledged that there may be some papers published prior to 1999 that may have been pertinent but not included in the review. The study was also limited to those papers from which clear implications for control of human swimming. Thus, papers were limited to those that had clear implications for control of cyclical actions in which there are complex frequency and phase characteristics between body parts that are stable as well as essential for optimal performance.

Other limitations are typical of systematic reviews. We chose two databases, but others, such as PubMed and Google Scholar, might have turned up different papers. Good, solid work that has not been published in peer-reviewed journals (the “desk drawer problem”) was, of course, not included, nor were papers that appeared only as book chapters or doctoral theses. A final limitation is that some papers that may have been relevant were not identified using our keyword choice.

## 5. Conclusions

This review has uncovered several concepts of neural control that could be applied to human butterfly and front crawl swimming to explain how complex rhythms might be achieved to optimize performance. In the absence of direct evidence from human swimmers while swimming, it must be emphasized that the review has given rise to only hypothetical explanations. Nevertheless, these provide a stimulus for empirical research to test the possibilities identified.

A common theme emerging from the review was that control of movement is simplified by innervating muscle synergies by way of spinal CPGs. These typically behave like oscillators enabling stable repetition across cycles of movements. Mathematical and computer modelling has shown that movements can be produced that closely resemble the actual rhythmical or wavelike motion of the body and limb movement of species ranging from arthropods to humans with a parsimonious number of control parameters. Parsimony of control is also achieved by innervation of muscle collectives or synergies. Modulation of sequencing of CPGs, signal strength, inhibition or excitation of centers in response to sensory feedback, in combination with input from the higher centers, provides the flexibility to cope with perturbations, to change speed or direction, and to allow turning.

Unfortunately, the search did not yield studies of movements that matched the complexity of rhythms observed in butterfly and front crawl swimming in which precise phase relationships among motions with different cycle frequencies are required for optimal performance. However, the two-layer model proposed by Saltiel et al. [35] to explain locomotion in cats might also facilitate the appropriate phase relationship among the cyclical motions of the various body parts in human swimming. The “pattern formation” layer of the CPGs would innervate muscle synergies in bursts according to a “temporal grid” while a “travelling wave layer” would maintain learnt optimal phase relationships among the cyclic actions which are operating at integer multiples of the lowest frequency in swimming, i.e., the two-beat body wave in butterfly and the two-beat body roll in front crawl. In keeping with the empirical evidence arising from the myriad in-vivo, fictive, in-vitro, and modelling of neural control in other species, flexibility to adapt to constraints, such as the physiological constraints associated with race distance, could arise from modulation of the CPG system in response to sensory and proprioceptive feedback and input from higher centers.

To better understand the neurological processes involved in becoming skilled in complex rhythmical activities such as butterfly and front crawl swimming, we propose a two-pronged approach. First, we need to analyze the changes in phase, and relative phase, of the composite body actions that occur during the period of skill acquisition from novice to proficient performance. The changes in rhythmic coordination must also be linked to the changes in the variables to be optimised—swimming speed, energetic efficiency, and hydrodynamic variables, including the effectiveness of generating propulsion and minimizing fluid resistance. Second, we need to investigate changes in the nervous system manifest in electroencephalographic (EEG) brainwave signals, spinal signals, and supplement those data with electromyographic (EMG) data to determine the relationship between muscle innervation and the efferent signals at spinal and cortical levels.

## Figures and Tables

**Figure 1 brainsci-10-00215-f001:**
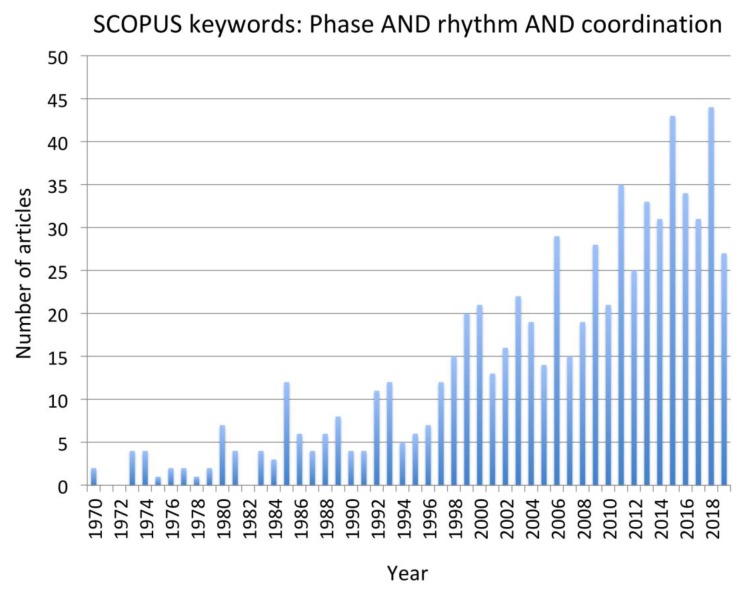
Frequency of publications matching the keywords “phase AND rhythm and coordination” in SCOPUS and Web of Science databases from 1979 to 2019.

**Figure 2 brainsci-10-00215-f002:**
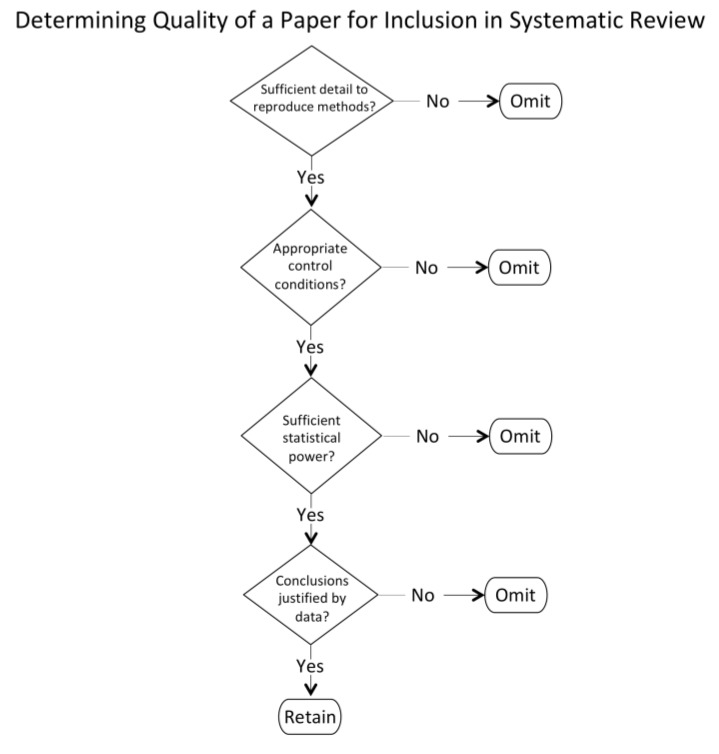
Flowchart for assessing the quality of papers.

**Figure 3 brainsci-10-00215-f003:**
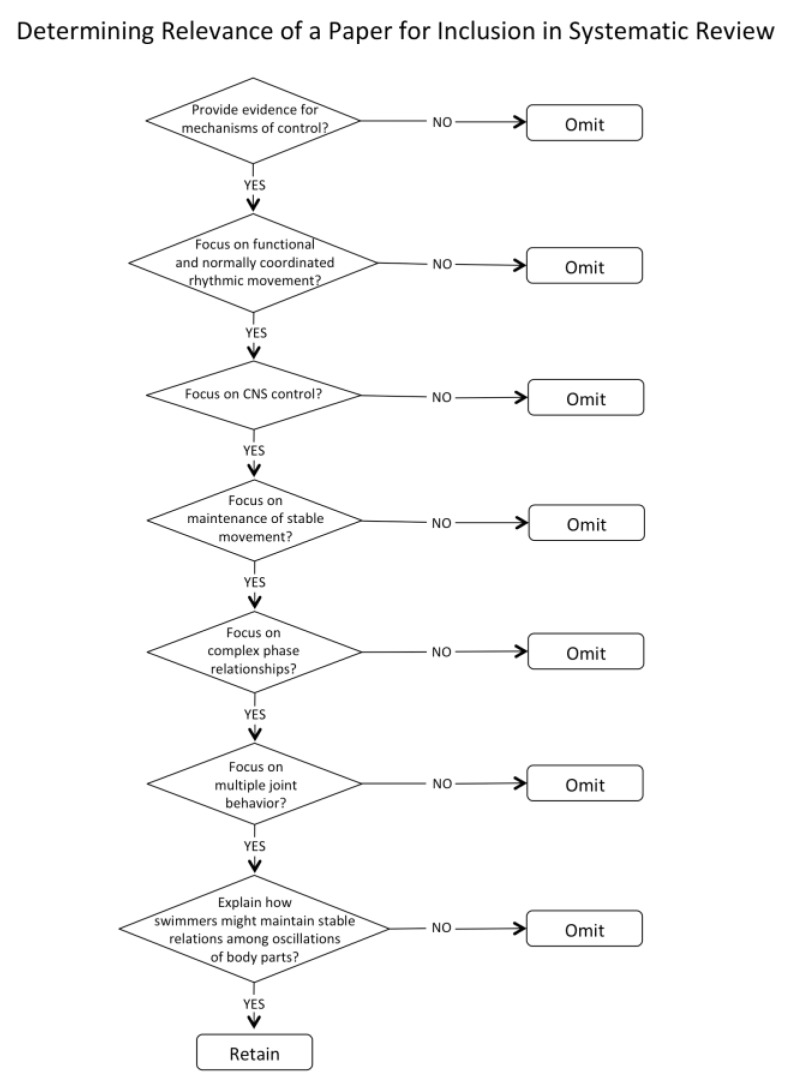
Flowchart for assessing the relevance of papers.

**Figure 4 brainsci-10-00215-f004:**
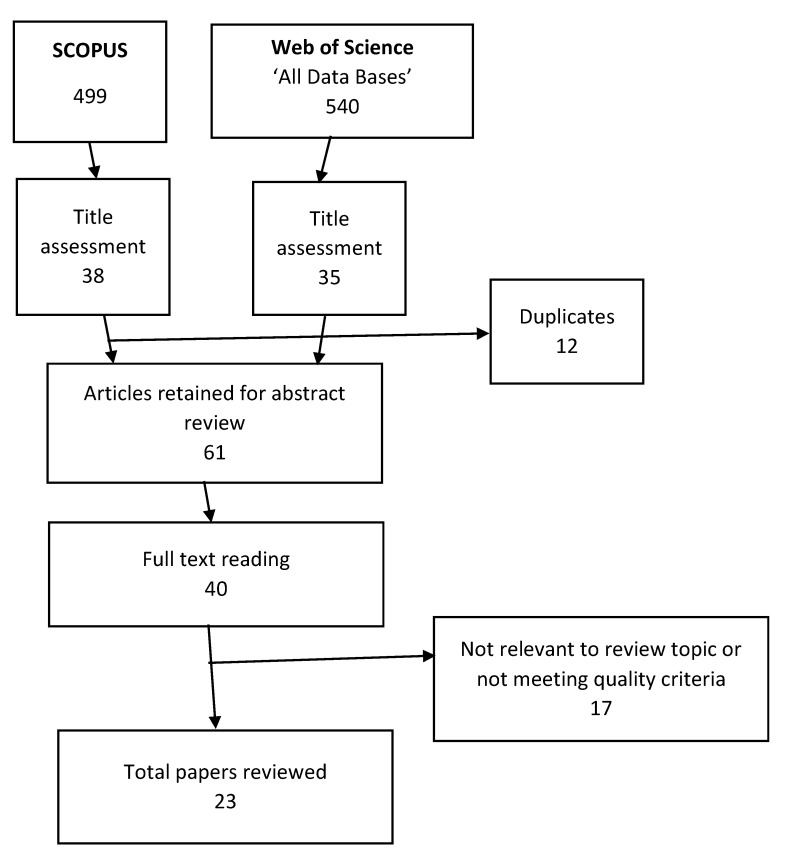
Flowchart of the systematic literature search based on the inclusion and exclusion criteria.

**Figure 5 brainsci-10-00215-f005:**
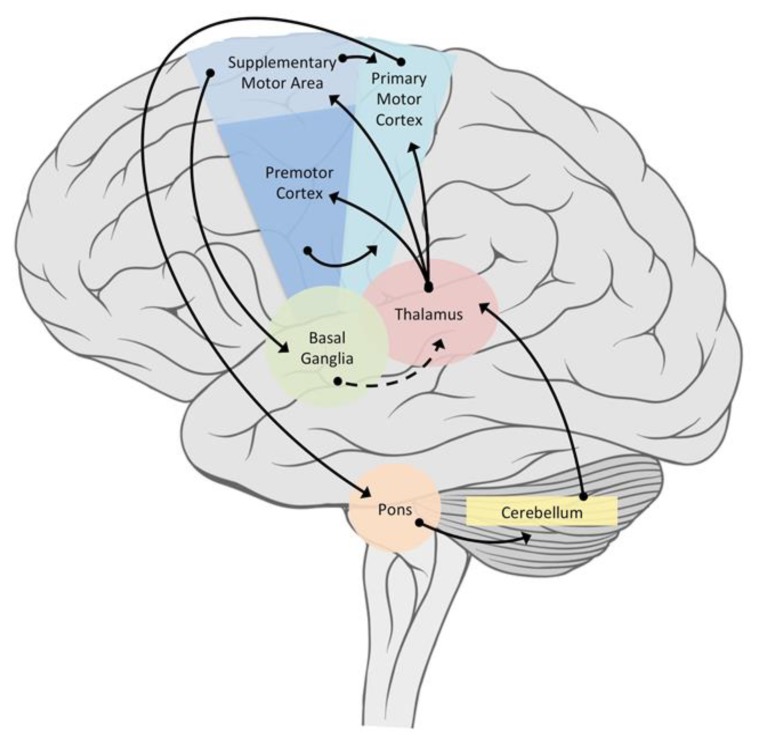
Connectivity in the brain among principal areas responsible for the motor control and coordination of complex rhythmic activities. Solid line = excitatory projections. Dashed line = inhibitory projections. Note that PFC and some parietal areas (not shown) are also important for maintaining different coordination patterns (goals and sensory monitoring) to determine if goals are being maintained.

**Table 1 brainsci-10-00215-t001:** Mean phase differences and velocities of the two-beat wave travel between body landmarks in the Sanders et al. [11] study.

	Mean Phase Difference (Degrees)		Mean Wave Velocity Relative to the Body (m/s)		Correlation ^1^	
Body Landmark	Males	Females	Males	Females	Males	Females
Vertex–shoulder	35	31	2.2	2.0	−0.09	0.18
Shoulder–hip	143	136	1.5	1.2	0.56	0.36
Hip–knee	44	60	1.8	2.2	0.47	0.46
Knee–ankle	26	46	3.8	2.1	0.77	0.77
Vertex–ankle	248	247	1.9	1.6	0.88	0.96

^1^ Correlation between the velocity of the wave travel and the center of mass velocity.

**Table 2 brainsci-10-00215-t002:** Phase (degrees) of the two-beat travelling wave and the four-beat travelling wave of a typical national level butterfly swimmer for the oscillations of the hip, knee, and ankle ([13] Sanders, 2007).

	Two Beat Wave (Degrees)	Four Beat Wave (Degrees)
Hip	201	89
Knee	266	136
Ankle	323	204

**Table 3 brainsci-10-00215-t003:** Mean body wave velocities of the flutter kick obtained in the Sanders [13] study of three levels of learners and a group of skilled swimmers.

	Hip–Knee Wave Velocity	Knee–Ankle Wave Velocity
Level 1	8.2	2.5
Level 2	8.3	4.1
Level 3	7.3	3.8
Skilled	2.8	3.2

**Table 4 brainsci-10-00215-t004:** Summary of articles screened in the systematic search listed in chronological order.

Author/Year	Purpose of Study (Abridged)	Data Sources	Species Studied/Modelled	Journal	Impact Factor
Calvitti and Beer (2000) [17]	To begin a systematic analysis of a distributed model of leg coordination	Computer model simulation of coupled leg oscillators	Stick insect Carousius Morosis	Biological Cybernetics	1.96
Saltzman and Byrd (2000) [18]	To explore the hypothesis that intergestural phasing relationships are implemented via coupling terms in a non-linear dynamical systems model	Computer model of coupled oscillators controlling speech	Humans	Human Movement Science	1.93
Dhamala et al. (2002) [19]	To study the neural correlates of rhythmic finger tapping	fMRI of brain activity	Humans	NeuroImage	5.81
Sternad and Dean (2003) [20]	To investigate the coupling effects in discrete and rhythmic movements	Upper limb kinematics and kinetics; EMG	Humans	Human Movement Science	1.93
Van Emmerik, Hamill, and McDermott (2005) [21]	To provide an overview of the empirical evidence for the functional role of variability in the stability and adaptability of human gait.	Phase relationships from kinematics of human gait	Humans	Quest	1.82
Ford Wagenaar and Newell (2007) [22]	To investigate the effects of auditory rhythms and arm movement on inter-segmental coordination during walking in persons who have suffered a stroke	Phase relation between upper and lower body segment kinematics	Humans	Gait and Posture	2.41
Drew, Kalaska, and Krouchev (2008) [23]	To address the functions of the motor cortex in control of gait	Review	Various	Journal of Physiology	5.04
Kozlov et al.(2009) [24]	To demonstrate general control principles that can adapt the Lamprey CPG network to different demands.	Computer model of the Lamprey CPG	Lamprey	PNAS	9.58
Pitti, Niiyama and Kuniyoshi (2010) (31)	to implement neuromodulators that can regulate the coordination between the body and the controllers’ dynamics to different gait patterns, either oscillatory or discrete.	Robotic elbow and leg system with Neuromodulators of CPGs	Vertebrates	Autonomous Robots	3.63
Ledberg and Robbe (2011) [25]	to investigate if and how the hippocampal theta rhythm is influenced by the periodic movements of locomotion.	Theta rhythms of Hippocampus and kinematic oscillations of the head	Rats	PLoS ONE	2.78
Snapp-Childs, Wilson, Bingham (2011) [26]	To test the hypotheses of the Bingham Model relating to stability of relative phase	Kinematics of a joystick task with 180 degrees relative phase at different oscillation frequencies	Humans	Experimental brain research	1.88
Thibaudier et al. (2013) [27]	To evaluate cycle and phase durations and footfall patterns of cats to assess directional control of fore and hind limbs	Frequency and phase durations of fore and hindlimb kinematics across speeds of split treadmill.	Cats	Neuroscience	3.24
Zhang et al. (2014) [28]	To understand how biologically salient motor behaviours emerge from properties of the underlying neural circuits.	Computational fluid dynamics; neural model of CPGs	Crayfish	PNAS	9.58
Ryczko (2015) [29]	To precisely define the different axial patterns underlying the different forms of locomotion in vivo.	Video-based kinematics and indwelling EMG	Salamanda	Neurophysiol	2.59
Harischandra, Krause, and Durr (2015) [30]	To introduce a general modelling framework of Central Pattern Generators (CPGs) for tactile exploration behaviour	CPG models with phase coupled Hopf oscillators	Stick insect	Frontiers in computational neuroscience	2.32
Hunt et al. (2015) [31]	To develop a model to explore the difference in phase timing in trotting rats.	Neural control model controlling 14 joints with Hill muscles	Rats	Bioinspiration and Biomimetics	3.13
Danner et al. (2016) [32]	To develop a computational model of spinal circuits to explain phase changes and gait transitions	Spinal circuit computer model with four rhythm generators and commissural excitation/inhibition	Mice	Journal of Physiology	5.04
Amado et al. (2016) [33]	To investigate the integration of bimanual rhythmic movements and posture in expert marching percussionists.	Video-based kinematics of drumming. Dynamic center of pressure from force plate.	Humans	Human Movement Science	1.93
Chen et al. (2017) [34]	To investigate the intra- and inter-limb muscle coordination mechanism of human hands-and-knees crawling by means of muscle synergy analysis	EMG of forelimbs and hindlimbs. Muscle synergy analysis.	Humans	Entropy	2.42
Saltiel et al. (2017) [35]	To compare CPG function and the travelling wave in locomotion of frogs and cats.	Review	Frogs and Cats	Frontiers in Neural Circuits	2.28
Spardy and Lewis (2018) [36]	To investigate the role of long-range coupling in crayfish swimmeret phase-locking	Computer model including neural circuits beyond nearest neighbour	Crayfish	Biological cybernetics	1.96
Qi et al. (2019) [37]	To evaluate whether two different, independent rhythms that involved finger tapping and walking could be produced.	Force sensors and metronomes	Humans	Sci Rep	4.01
Dutta et al. (2019) [38]	To generate a range of rhythmic gait patterns using a CPG network	Robot control system with hardware equivalents of biological structures—spinal cord CPGs, muscles, sensors, and brain centers.	Robots	Nature Communications	11.88

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
