# Peer review of "Towards an Understanding of Control of Complex Rhythmical “Wavelike” Coordination in Humans"

_brainsci, 2020, doi:10.3390/brainsci10040215_

Round 1

Reviewer 1 Report

Thank you for giving me the opportunity to review this valuable study. First of all I must say that this paper is of great quality and provides very relevant and necessary information from my humble point of view. This review study shows great aspects of great potential as a very rigorous, deep and adequate theoretical framework that perfectly introduces the study as well as a good argumentation of why there is a need to carry it out.

I appreciate that meta-databases are used directly (Scopus/WOS), instead of choosing 2 or 3 databases without having a solid criterion. In addition to this, I think that another very appropriate aspect is to impose a cut-off or quality criterion. I would not want the latter to be misunderstood. Papers published in peer-reviewed journals with less than 1.5 need not be bad papers (all researchers have some in those journals), but there are actually papers published in that range that would need an application of some methodological rigour…

That said, for me this study needs little more to be published, but I would like to ask some questions to try to make the methodology as robust as possible (minor revision).

In the methods there comes a moment when the authors comment that there are articles before the year 2000 that serve to introduce but above all to discuss this review. Why do the authors make the cut in that year, don't the authors think that they are losing information by cutting in that year, why don't they try to make a search since 1950 (for example)? Or... What is the reason for cutting in 2000?

2.- Then, there is something that worries me at a methodological level. I think this is what is most important. In order to choose the articles, the first author determined which one was chosen and which one did not enter the review. Well, however this is a systematic review and therefore I myself (this is an example) from my home, with these search criteria should find the same papers as the authors of this review.

I am very concerned that another researcher (in a hypothetical case of replication) include other studies that the first author rejected. Do you think that you could provide a kind of table or appendix that would set out criteria used by the author to make the study 100% replicable by other research groups?

I think it would have been ideal to do this process of including the articles independently among the 2 authors of the study to see the concordance between them with a simple kappa index and to register reasons for exclusion or inclusion, let's say "better replicability".

3.- In general, this is a point of union of point 1 and 2. The methods must be slightly improved because we are in front of a systematic review.

4.- Please reference the papers in the table so that the reader can quickly access the reference section.

5.- Authors should put a "Limitations" section in the discussion.

6.- The discussion is correct and provides a lot of information. Perhaps too much. It would be convenient to shorten it slightly to facilitate the reading of the study. But in quality it is impeccable.

7.- There is some reference with underlining

8.- Be careful with the numbering of the sheets. When creating a horizontal sheet, the numbering order has been lost.

Author Response

Thank you for giving me the opportunity to review this valuable study. First of all I must say that this paper is of great quality and provides very relevant and necessary information from my humble point of view. This review study shows great aspects of great potential as a very rigorous, deep and adequate theoretical framework that perfectly introduces the study as well as a good argumentation of why there is a need to carry it out.

Thank you for this positive assessment.

I appreciate that meta-databases are used directly (Scopus/WOS), instead of choosing 2 or 3 databases without having a solid criterion. In addition to this, I think that another very appropriate aspect is to impose a cut-off or quality criterion. I would not want the latter to be misunderstood. Papers published in peer-reviewed journals with less than 1.5 need not be bad papers (all researchers have some in those journals), but there are actually papers published in that range that would need an application of some methodological rigour…

We agree that papers in journals with impact factors less than 1.5 might still be good papers. As it turned out there were not any relevant papers in such journals that were excluded solely on that criterion. Nevertheless, we thought it beneficial to show the impact factors in the table to indicate that all selected papers were published in moderate to high impact journals. In view of this comment we have now indicated that only one paper that made it through the screening based on title and abstract was not included due to its ultimate lack of relevance rather than its impact factor. Please see the rewording of the section in the method in response to your other comment below.

That said, for me this study needs little more to be published, but I would like to ask some questions to try to make the methodology as robust as possible (minor revision).

Thank you. Responses to the specific comments are provided below.

In the methods there comes a moment when the authors comment that there are articles before the year 2000 that serve to introduce but above all to discuss this review. Why do the authors make the cut in that year, don't the authors think that they are losing information by cutting in that year, why don't they try to make a search since 1950 (for example)? Or... What is the reason for cutting in 2000?

Yes, a very pertinent question. In conducting the initial search it was apparent that there had been rapid development of methods and sophistication of the mathematical models and exploratory techniques. Therefore we limited the eventual strategic approach to the last 20 years. Although there are important papers earlier that were foundational to the recent development we did not find any that were unique in their contribution to the focus of this paper.  However, some papers that were pre 2000 did provide models on which subsequent papers were based. The more recent papers had cited those but had also improved the models. We acknowledged those in the write-up and cited them showing their important contribution as foundational in the recent development. Therefore, we feel that this approach has captured the foundational work and acknowledged its contribution appropriately. We have now elaborated the rationale in the text body. We have also further explored the volume of papers arising from the initial keyword search and have found that for both data bases there becomes a ‘critical mass’ of over 20 papers per year from the year 1999 Therefore we have extended the search to include 1999 We have shown those frequency graphs as additional justification of the cut off at 1999.  The following wording has been added to the method section:

‘Given the rapid advancement in experimental approaches and mathematical models of CPGs papers the period 1999 to the time of writing (December 31, 2019) were reviewed. However, pertinent earlier articles that were cited as foundational to the recent work were included when necessary in the introduction or during the interpretation of the contribution of each paper in the Discussion. Also from around the year 1999 there has been a consistent ‘critical mass’ of more than 20 papers per annum identified through the chosen keywords (Figure 1).   The selected papers were all peer reviewed journal articles with impact factors over 1.5. Conference proceedings papers and dissertations were not considered.’

Following your later suggestion to include a section on limitations (see below).

2.- Then, there is something that worries me at a methodological level. I think this is what is most important. In order to choose the articles, the first author determined which one was chosen and which one did not enter the review. Well, however this is a systematic review and therefore I myself (this is an example) from my home, with these search criteria should find the same papers as the authors of this review.

I am very concerned that another researcher (in a hypothetical case of replication) include other studies that the first author rejected. Do you think that you could provide a kind of table or appendix that would set out criteria used by the author to make the study 100% replicable by other research groups?

We have now added more detail about the filtering process as below and in separate flowcharts for relevance and quality flowchart added to the document as figures. :

‘Two categories of criteria were applied in filtering the papers identified by the keyword search – quality and relevance. To meet the quality criteria a paper needed to report empirical data with rigor of methods and replicability evident from detail of the experimental procedures, as determined by the first author and checked by the second author in accordance with the criteria shown in Figure 2.  Given that the range of experimental approaches included in vivo, in vitro, and mathematical modelling, those general, rather than specific, criteria were applied to ensure quality.

To be considered relevant a paper needed to provide evidence for mechanisms of control that could contribute to explaining how swimmers might maintain stable relationships among oscillations of body parts in swimming. The criteria for inclusion are shown in Figure 3. The process was conducted sequentially by assessing relevance by title, then by abstract, and then by reading the full papers to further assess relevance and quality.

Only one paper that had progressed to reading of the full paper was deemed low quality. That paper also had limited relevance. Many rigorous papers were filtered during either the abstract assessment (17) or during the reading of the full text (14). Common reasons were: 1. Focus on methods of understanding or treating dysfunctional rhythmical movement rather than functional and normally coordinated rhythmical movement 2. Focus on output behavior or EMG rather than the central nervous system control 3. Focus on instability rather than maintenance of stable and sustainable movement. 4. Focus on entrainment of oscillatory behavior rather than maintenance of phase relationships other than in-phase or 180 degrees out of phase. 5. Focus on single joint behavior or coupling of a limited number of joints (often with an injury focus). The papers that were filtered out were checked by the second author to ensure agreement with regard to inadequate relevance.’

I think it would have been ideal to do this process of including the articles independently among the 2 authors of the study to see the concordance between them with a simple kappa index and to register reasons for exclusion or inclusion, let's say "better replicability".

The second author has now checked the rejected papers for relevance. That statement is now included in the addition highlighted above.

3.- In general, this is a point of union of point 1 and 2. The methods must be slightly improved because we are in front of a systematic review.

4.- Please reference the papers in the table so that the reader can quickly access the reference section.

We have now included numbers in the table corresponding to the numbers in the reference list.

5.- Authors should put a "Limitations" section in the discussion.

A ‘Limitations’ section has been added as suggested.

‘The review was limited to papers from 1999 to present for the reasons outlined in the method section. Although some foundational papers published prior to that period are identified when discussing the selected papers, it is acknowledged that there may be some papers published prior to 1999 that may have been pertinent but not included in the review. The study was also limited to those papers from which clear implications for control of human swimming. Thus, papers were limited to those that had clear implications for control of cyclical actions in which there are complex frequency and phase characteristics between body parts that are stable as well as essential for optimal performance.

Other limitations are typical of systematic reviews. We chose two databases, but others, such as PubMed, and Google Scholar, might have turned up different papers. Good, solid work that has not been published in peer reviewed journals (the desk drawer problem) was of course not included, nor were papers that appeared only as book chapters or doctoral theses. A final limitation is that some papers that may have been relevant were not identified using our keyword choice. ‘

6.- The discussion is correct and provides a lot of information. Perhaps too much. It would be convenient to shorten it slightly to facilitate the reading of the study. But in quality it is impeccable.

Thank you. We have checked the discussion to ensure that there is both concise and has sufficient information provided to establish the points clearly.

7.- There is some reference with underlining

Underlining has now been removed.

8.- Be careful with the numbering of the sheets. When creating a horizontal sheet, the numbering order has been lost.

Checked and corrected as necessary.

Reviewer 2 Report

Overall this is a well-written review with methodologically literature search and discussion. The conclusion provides direction to novel hypotheses to dissect out the relationship between cyclical motions and activity.

Author Response

Overall this is a well-written review with methodologically literature search and discussion. The conclusion provides direction to novel hypotheses to dissect out the relationship between cyclical motions and activity.

Thank you for this positive assessment.

Reviewer 3 Report

This is an excellent paper dealing with Control of Complex Rhythmical ‘Wavelike’ Coordination in Humans.

You have submitted quite a fascinating manuscript and I sure would like to see it printed soon.

This paper appears to be reporting some significant points, however, the impact is lost by a limited tables.

Although I have no doubt about the quality of the presented work, I recommend to revise the figure/table so that the purpose appears more easily understanded.

1. In Table 2, I think it is good to put "degree" as a unit under two items (Two Beat Wave / Four Beat Wave).

2. I suggest it is good to insert the figure of the relationship between brain activity and the rate and complexity of the rhythms. The simple figure about the circuit of the primary motor cortex, premotor cortex, auditory cortex, basal ganglia, thalamus, and the cerebellum are helpful for reader's under standing your logic.

Overall, I think this paper would be acceptable for publication if the above points are revised.

Author Response

This is an excellent paper dealing with Control of Complex Rhythmical ‘Wavelike’ Coordination in Humans.

You have submitted quite a fascinating manuscript and I sure would like to see it printed soon.

This paper appears to be reporting some significant points, however, the impact is lost by a limited tables.

Although I have no doubt about the quality of the presented work, I recommend to revise the figure/table so that the purpose appears more easily understanded.

  1. In Table 2, I think it is good to put "degree" as a unit under two items (Two Beat Wave / Four Beat Wave).

‘Degrees’ added as suggested.

  1. I suggest it is good to insert the figure of the relationship between brain activity and the rate and complexity of the rhythms. The simple figure about the circuit of the primary motor cortex, premotor cortex, auditory cortex, basal ganglia, thalamus, and the cerebellum are helpful for reader's under standing your logic.

We have now added a figure as suggested.

Figure 3 (?).Connectivity in the brain among principal areas responsible for the motor control and coordination of complex rhythmic activities. Solid line = excitatory projections. Dashed line = inhibitory projections. Note that PFC and some parietal areas (not shown) are also important for maintaining different coordination patterns (goals and sensory monitoring to determine if goals are being maintained.

Overall, I think this paper would be acceptable for publication if the above points are revised.

Thank you for your favourable review and suggestions